# Addition of *Mentha arvensis* in Infusions of *Cleistocalyx operculatus* Improves the Hedonic Score and Retains the High Antioxidant and Anti Lipid-Peroxidation Effects

**Tran Thi Le Minh [1,\*], Luu Thi Bich Kieu [1], Son Thi Tuyet Mai [1,2], Dang Long Bao Ngoc [1], Le Thi Bich Thuy [1], Nguyen Thi Quyen [1], Ton Trang Anh [1], Le Van Huy [1], Nguyen Vu Phong [1], Chung Thi My Duyen [3], Nguyen Hoang Minh [3] and Gontier Eric [4,\*]**

[1] Faculty of Biological Sciences, Nong Lam University, Ho Chi Minh City 70000, Vietnam
[2] Orient Foods International Joint Stock Company, 263A, Phan Trung Street, Tan Tien Ward, Bien Hoa 76100, Vietnam
[3] Research Center of Ginseng & Material Medical, 41 Dinh Tien Hoang, Ben Nghe Ward, District 1, Ho Chi Minh 70000, Vietnam
[4] UMR Transfrontalière BioEcoAgro INRAE-1158, Laboratoire BIOPI-UPJV, Université de Picardie Jules Verne, UFR Sciences, 33 rue Saint Leu, 80000 Amiens, France
[\*] Correspondence: ttlminh@hcmuaf.edu.vn (T.T.L.M.); eric.gontier@u-picardie.fr (G.E.)

**Abstract:** (1) Background: Many human diseases are associated with oxidative stress, which is caused by reactive oxygen species and free radicals generated in living cells. Some biomass extracts derived from various types of plants can act as efficient drugs against pathological disorders related to oxidative stress. Numerous herbal blends have thus been shown to improve health. *Cleistocalyx operculatus* (Roxb.) Merr. and L.M.Perry teas have been considered in that way. Problem: Because of amertume, the taste of *C. operculatus* avoids or limits a large use of such alleged healthy leaf infusions. (2) Methods: The phytochemistry, oxygen, free radical scavenging activity, and antilipid peroxidation of *C. operculatus* teas were here studied in vitro. Then different mixes of *C. operculatus* and *Mentha arvensis* were infused together and tasted in a hedonic test. The chemical and biological properties of the best mix were then analyzed. (3) Results: The herbal blend of *C. operculatus* revealed significant scavenging effects on 1,1-diphenyl-2-picrylhydrazyl radical (DPPH) with $IC_{50}$ values of 35.6 µg/mL. Induced by hydroxyl radicals, this beverage could significantly inhibit the lipid peroxidation of mouse brain homogenates. Our results demonstrate that the lipid peroxidation inhibition of dried leaves of *C. operculatus* might be related to its scavenging effects on oxygen free radicals. This *C. operculatus* bitter blend was then combined with various amounts of *M. arvensis* Linn. The taste was evaluated, and further biochemical analyses were conducted on the best 7/3 ratio. They showed that the healthy properties were maintained. (4) Conclusion: The optimized 7/3 ratio of the *Cleistocalyx/Mentha* mix led to the best results in terms of taste (sensory tests). It is demonstrated that the potential health benefits against oxidative stress remained high as compared with pure *C. operculatus* infusion.

**Keywords:** *Cleistocalyx operculatus*; *Mentha arvensis*; antioxidant activity; phytochemicals; reactive oxygen species; free radicals; lipid peroxidation

## 1. Introduction

*Cleistocalyx operculatus* (Roxb.) Merr. and L.M.Perry plant is widely distributed in tropical countries and is commonly used as a source of phenolic and flavonoid compounds. Anthocyanins are antioxidants that are found in the buds and fruits of *C. operculatus* leaves and are used to treat digestive conditions [1]. A recent study has also revealed the antiaging properties of *C. operculatus* [2]. The plant's flower bud extract has antihyperglycemic activity due to its inhibitor activity against the α-glucosidase enzyme [3]. Its fruit extract has been shown to exert neuroprotective effects and decrease the rate of hepatocarcinogenesis [4,5].

As inferred from the literature, the most active ingredients were obtained from crude extracts involving one of the four possible extracts/fractions (Table 1) corresponding to different mixtures of pure solvents (ethanol, methanol, hexane, ethyl acetate, chloroform, dichloromethane, and water), pure water, or pure chloroform. The heating or ultrasonic-assisted process was shown to reduce the extraction time. The bioactive compounds and antioxidant activity of (young and old) leaves, fruits, and buds were identified and evaluated by different research groups (Table 1).

**Table 1.** Extraction methods described for *C. operculatus* phytochemical studies.

| Methods | Solvents | Process | Sample | Bibliography |
|---|---|---|---|---|
| Crude extract | Water, methanol, chloroform, ethanol | Boiling: 100/65/60 °C + sonication | Fruits, leaves | [4,6] |
| Fraction | Methanol/water/hexane/EtOAc/CH$_2$Cl$_2$ | Sephadex LH 20 silica gel | Buds | [7] |
| Oil leaf extract | Water Chloroform | Hydrodistillation Likens–Nickerson | Leaves | [8,9] |

The use of natural herbal extracts for improving the (i) sensory properties and (ii) health benefits of *C. operculatus* infusions has been claimed for some time. Nevertheless, scientific lines of evidence need to be obtained to objectify the real health benefits of this bitter infusion that subsequently needs taste improvement. Mint leaves can be infused alone or mixed with other medicinal leaves for the infusion. Mint has also been long known as a medicinal plant. Moreover, mint plants contain menthol, which provides a spicy, cooling, and refreshing flavor. Menthol is the main compound of the *Mentha arvensis* L. essential oil. Its level was found to be 77% in such extracts [10]. Mint oil can (i) increase the ambulatory activity of mice and (ii) stimulate the central nervous system [11]. It is used to relieve pain spasms and arthritic problems. In addition, mint oil has several other potential benefits; it has anti-inflammatory, analgesic, anti-infectious, antimicrobial, antiseptic, antispasmodic, astringent, digestive, carminative, and fungicidal effects, as well as acting as a vasoconstrictor and decongestant with stomachic properties [12].

Infusions of *C. operculatus* leaves or flower buds have been commonly used for the treatment of influenza and some digestive conditions [1] and have also been used as components of herbal blends that can be combined with different plant sources with valuable secondary metabolites such as phenolics, flavonoids, alkaloids, carotenoids, saponins, and terpenoids [13]. Nevertheless, a natural amertume limits the appetence for the bitter infusions of *C. operculatus*. No study has been reported on the inhibitory effect of *C. operculatus* bitter infusions, including the combination of *C. operculatus* and *M. arvensis* dried leaves, against lipid peroxidation in the mouse brain.

In this study, a combination of *C. operculatus* and *M. arvensis* leaves was studied in order to exploit the potential use of the resulting mixed infusions as functional food drinks. In the herbal blend tested, the total flavonoid and terpenoid concentrations were determined. Sensory tests were also performed to improve the bitter taste of the infusions. The DPPH radical scavenging activity and the lipid peroxidation inhibitory activity were then determined on the most appetizing ratio of *C. operculatus* and *M. arvensis*.

## 2. Materials and methods

### 2.1. Animals

Male *Swiss albino* mice (25 ± 2 g, aged 5–6 weeks) were supplied by the Institute of Vaccines and Medical Biological Products in Nha Trang City. Before experimentation, all the animals were kept for 1 week under the same laboratory conditions in terms of temperature (22 ± 2 °C), relative humidity (70 ± 4%), and a 12 h light/dark cycle; they received a nutritionally standard diet and tap water.

Our research was approved by the National Vietnamese authorities with the approval No. 20210108NLU.

## 2.2. Plant Material and Extraction

Fresh leaves of *Cleistocalyx operculatus* (Roxb.) Merr. and L.M.Perry and *M. arvensis* were collected from Ho Chi Minh City in 2021. They were washed and air-dried at room temperature ($25 \pm 2$ °C) for 1 day and were then fully dried in a drying oven (ISUZU SSSF 214S) at 80 °C, ground into powder, and stored in an airtight container in a cool place. Based on the literature (Table 1), further analyses were performed.

## 2.3. Preliminary Phytochemical Screening

Qualitative phytochemical screening of *C. operculatus* was performed in order to detect the presence of essential phytoconstituents, such as alkaloids, saponins, flavonoids, terpenoids, and phenols. The dried and powdered leaves of *C. operculatus* (1 g) were extracted with 15 mL of either methanol, ethanol, acetone, or chloroform for 10 min.

Alkaloids screening: Briefly, 3 mL of the solvent extract was dissolved in 3 mL of 1% HCl and treated with Wagner's reagent, which formed a brown/reddish brown precipitate indicative of the presence of alkaloids [14].

Flavonoids screening: Briefly, 1 mL of the solvent extract was dissolved in 0.3 mL $NaNO_2$ 15%; then, 5 min later, 0.3 mL AlCl3 10% was added. After another 5 min, 4 mL of NaOH 4% was added. The tubes were allowed to stand for 1 min. The formation of an orange color indicated the presence of flavonoids [2].

Phenols screening: Briefly, 2 mL of the solvent extract was added to a few drops of 5% $FeCl_3$. The resulting bluish-black color confirmed the presence of phenols [15].

Saponins screening: Five mL of each solvent extract was evaporated until dried and then added to 5 mL of distilled water and vortexed for 15 s. The formation of foam indicated the presence of saponin, as previously described [14].

Terpenoids screening: Briefly, 5 mL of the extract was mixed with 2 mL of chloroform and 3 mL of concentrated $H_2SO_4$ (95%). The formation of a reddish-brown color in the inner face indicated the presence of terpenoids [16].

## 2.4. Quantification of Flavonoid and Terpenoid Content

The dried and powdered leaves of *C. operculatus* (1 g) were extracted with 15 mL of boiling water for 10 min.

### 2.4.1. Quantification of Flavonoid Content

The total flavonoid contents of the extracts were determined using the colorimetric method of Manosroi et al. (2015) [2]. Briefly, 1 mL of the extract was mixed with 0.3 mL of 15% ($w/v$) $NaNO_2$ and 0.3 mL of 10% ($w/v$) $AlCl_3$. After 5 min, 4 mL of 4% ($w/v$) NaOH was added. After 1 min, water was immediately added to adjust the final volume to 10 mL. Then, the mixture was thoroughly mixed by shaking for 15 min at room temperature ($27 \pm 2$ °C). Solution absorbance was measured at 510 nm (Perkin Elmer MBA 2000). The total flavonoid content of the samples was expressed as milligrams of quercetin equivalents (mg QE/g) of the dried extracts and calculated with the linear equation based on the calibration of pure quercetin. All measurements were performed in triplicate.

### 2.4.2. Quantification of Terpenoid Content

A vanillin-sulphuric acid assay [16] was performed to determine the terpenoid content of the plant materials by incubating 1 mL of each sample with 0.5 mL of 8% ($w/v$) vanillin in ethanol and 0.5 mL of 72% ($v/v$) sulphuric acid in water for 15 min at 60 °C in a water bath while shaking. After cooling in water at ambient temperature for 5 min, the absorbance of the samples was measured at 560 nm (PerkinElmer MBA 2000). The standard stock solutions of menthol were prepared using ethanol. The terpenoid content of the samples was expressed in milligrams of the standard equivalents per gram of each plant sample (mg SE/g).

### 2.5. Mixing Herbal Remedies

Before oven drying, the leaves of *C. operculatus* and *M. arvensis* were incubated in a stainless-steel jar at 60 °C for 2 h to bleach the plant material (inhibition of enzymatic activities). The dried leaves were ground into coarse powder and mixed with varying proportions (Mix 1: 100% *C. operculatus*; Mix 2: 90% *C. operculatus* and 10% *M. arvensis*; Mix 3: 70% *C. operculatus* and 30% *M. arvensis*; Mix 4: 50% *C. operculatus* and 50% *M. arvensis*; and Mix 5: 100 % *M. arvensis*). The further determination of total flavonoid and total terpenoid concentrations was achieved through spectrophotometry, HPLC, and GC, as described below.

### 2.5.1. HPLC Determination of Quercetin

The herbal blend (1 g) was extracted with 25 mL of 70% ethanol ultrasonicated for 15 min. The filtrate was separated from the residue using Whatman filter paper and transferred to a 100 mL volumetric flask. The residue was then extracted with 25 mL of 70% ethanol ultrasonicated for 15 min. After filtration through the Whatman filter paper, the filtrate was transferred to the same 100 mL volumetric flask. This process was repeated two more times to complete the extraction. The flask was then filled to the mark with 70% ethanol. The sample was filtered through a nylon syringe filter with a 0.22 μm pore size before final analysis using HPLC. The analysis of quercetin in the extract was conducted using high-performance liquid chromatography (Agilent 1100) using a Hypersil BDS-C18 column (5 μm particle size, 250 mm × 4.6 mm). The isocratic mobile phase was acetonitrile:water:acetic acid (30:70:1) at a flow rate of 1 mL/min. The UV detector was fixed at 368 nm.

### 2.5.2. GCMS Determination of Menthol

The herbal blend (0.2 g) was extracted with 10 mL of n-hexane ultrasonicated for 15 min. The filtrate was separated from the residue using Whatman filter paper and transferred to a 50 mL volumetric flask. The residue was extracted with 10 mL of n-hexane ultrasonicated for 15 min. After filtration through the Whatman filter paper, the filtrate was transferred to the same 50 mL volumetric flask. This process was repeated two more times to complete the extraction. The flask was filled to the 50 mL mark with n-hexane. The samples were filtered through a PTFE syringe filter with a 0.22 μm pore size before the final analysis using GCMS.

The analysis of menthol in the powder extract was carried out using a gas chromatograph coupled with a mass spectrometer (GC/MS Agilent 6890N/5973) with a DB-5MS capillary column (30 m × 0.25 mm i.d., 0.25 μm, 5% phenyl-methylpolysiloxane; Agilent J&W). Helium was used as the carrier gas at a flow rate of 1 mL/min. The column temperature was initially programmed at 60 °C and then increased to 180 °C at 12 °C/min and held for 2 min. The injector and detector temperatures were 220 °C and 230 °C, respectively. The ionization energy was 70 eV. The compounds were identified based on a comparison of their mass spectra using the data in NIST 14 (National Institute of Standards and Technologies, Mass Spectra Libraries).

### 2.5.3. Sensory Analysis Using a 9-Point Hedonic Scale

Freshly boiled mineral water (100 mL as the reference or 150 mL as the improved concentration determined later) was poured onto 2 g of each herbal blend sample for 10 min. The sensory profiling of the developed beverage powder was carried out using a 9-point hedonic scale. There were 45 panel members through which the samples were rated for acceptability using the 9-point scale [17].

### 2.5.4. Effect of Brewing Time on Total Flavonoids and Terpenoids

Briefly, 2 g of the herbal blend was brewed using 100 mL of freshly boiled mineral water for 6, 10, and 60 min. The infusions were prepared in a small porcelain teapot, which was closed with a porcelain lid during brewing. Three samples of the herbal blend infusions

were prepared for each brewing condition for the analysis of flavonoids and terpenoids, and the mean of the results for the three samples was used.

### 2.6. Determination of DPPH Radical Scavenging Activity

Briefly, after lyophilization of infusion samples and further dilution in methanol, 0.5 mL of various concentrations of each sample (10, 50, 100, 250, 500, 750, and 1000 µg/mL) or ascorbic acid (0.5, 0.25, 0.1, 0.05, and 0.01 mM) were added into a tube containing 0.5 mL of a 0.6 mM DPPH solution dissolved in methanol, and the volume was uniformly increased to 4 mL using methanol. The solution was mixed and then allowed to stand in the dark at room temperature for 30 min. Absorbance was taken at 515 nm using methanol as the blank solution on a UV-visible spectrometer. Then, 0.5 mL of DPPH was added to 3.5 mL of methanol, and absorbance was taken for reading the control. All the analyses were run in triplicate.

### 2.7. Determination of Lipid Peroxidation Inhibitory Effects

After sampling mouse brain tissues, the tissues were immediately placed in ice-cold 5 mM phosphate buffer (pH 7.4) with the brain tissue and phosphate buffer ratio of 1:10 (*w*/*v*). The tissue was then homogenized at 13,000 rpm in ice-cold conditions. Then, 0.1 mL of various concentrations of each sample (as described above in 2.4) or Trolox (10, 5, 1, 0.5, and 0.1 mM) was added to 0.5 mL of brain homogenate, and the volume was increased to 2 mL using 50 mM phosphate buffer. The mixture was then incubated at 37 °C for 15 min to allow lipid peroxidation and, thus, MDA (malondialdehyde) production. The reaction was stopped with 1 mL of 10% trichloroacetic acid. The tube was then centrifuged at 19,000 g at 5 °C for 5 min. The supernatant (2 mL) was transferred into a different tube and then allowed to react with 0.8% TBA (thiobarbituric acid) solution (1 mL) at 95 °C for 15 min. The control was prepared as above without a test sample. The absorbance of the mixture was measured at 532 nm against a blank solution with no sample or TBA after the tubes reached room temperature.

## 3. Results and Discussion

### 3.1. Phytochemical Analysis of the Pure C. Operculatus Leaf Powder

#### 3.1.1. Phytochemical Screening

Qualitative phytochemical screening of *C. operculatus* was performed to detect the presence of essential phytoconstituents, such as alkaloids, saponins, flavonoids, terpenoids, and steroids (Table 2). As shown in Table 2, *C. operculatus* leaves contain various phytochemicals or polyphenols (such as flavonoids, saponin, phenols, terpenoids, and alkaloids), which are known to have potent antioxidant and/or free radical scavenging activities. Terpenoids are the main antioxidant compounds of essential oils. The mode of action underlying the plant's antioxidant properties may thus be mainly related to that family of compounds in essential oils (EOs); it may also be due to other chemicals (flavonoids, saponins, phenols, and alkaloids) that may contribute alone or with synergic effects to the antioxidant activities.

**Table 2.** Detection of phytochemical constituents in extracts from *C. operculatus* leaves, depending on the type of solvent used for the extraction procedure.

| Test | Methanol | Ethanol | Acetone | Chloroform |
|---|---|---|---|---|
| Flavonoids | + | + | + | − |
| Terpenoids | + | + | + | + |
| Alkaloids | − | − | + | − |
| Phenols | + | + | + | + |
| Saponins | − | + | − | − |

Note: + = present and − = absent. Each data point corresponds to triplicates.

3.1.2. Quantification of Total Flavonoids and Terpenoids

Flavonoids and terpenoids are the main groups of secondary metabolites found in natural products capable of antibacterial properties via multiple mechanisms of action [18,19]. Additionally, several recent reports have confirmed that the major classes of phytochemicals, including flavonoids and terpenoids, have significant antioxidant and free radical scavenging activities [20]. Thus, we determined the total flavonoid and terpenoid levels in the young/mature/old leaves (1–3/4–5/> 6 weeks old and 7–9/12–14/10–12 cm length, respectively) of 3-years-old *C. operculatus* plants. The powdered and dried *C. operculatus* was extracted with boiling water (Table 3), and the percentage yield was further calculated on a dried weight basis (after the extracts were kept at 4 °C in a capped, airtight container until use).

**Table 3.** Total flavonoid and terpenoid contents in *C. operculatus* leaves.

| Phytochemicals | Treatment | | |
|---|---|---|---|
| | Young Leaves | Mature Leaves | Old Leaves |
| Flavonoids (mg QE/g) | 2.34 [b] ± 0.15 | 1.76 [c] ± 0.04 | 2.73 [a] ± 0.21 |
| Terpenoids (mg SE/g) | 4.00 [c] ± 0.06 | 4.28 [b] ± 0.05 | 5.43 [a] ± 0.19 |

Means followed by a different letter [a–c] are significantly different at an alpha level of 0.05% according to a Least Significant Difference test.

The results of spectrophotometric determination for the total flavonoid and terpenoid contents are presented in Table 3. The determination of the total flavonoid content was based on the reaction of the complexation of flavonoids with $AlCl_3$ and absorbance measurement at 510 nm [2]. The spectrophotometric quantification of the total terpenoid content was carried out using a vanillin-sulphuric acid assay with absorbance at 560 nm [16]. Means with no letter in common indicate significant differences between treatments ($p \leq 0.05$).

The results of Table 3 demonstrate the percentage yields of the young, mature, and old leaves extracted with boiling water. For flavonoids, quercetin was used as the reference for semi-quantitative determination. The percentage yield extracted from old leaves is 2.7 mg QE/g (milligrams of quercetin equivalent per gram of the dry matter extracted). For terpenoids, menthol was used as the quantitative reference. The terpenoid yield in the old leaves' extracts is 5.4 mg SE/g (milligrams of sesquiterpenes based on menthol used for relative quantification and expressed as equivalent per gram of the dry matter extracted). The flavonoid and terpenoid contents in the old leaves' extracts are higher than those measured for young and mature leaves. Based on Wang et al. [21], we can speculate that the young and mature leaves suffered less damage and thus contained fewer secondary metabolites than the older leaves, which could accumulate more defense compounds during their lifetime. Effectively, many secondary metabolites found in plants are involved in defense against herbivores, pests, and pathogens. Plant responses to herbivory with traits related to both defense and tolerance are thus affected by plant age [21]. Older plants are capable of maintaining stronger chemical defense mechanisms than young plants, possibly because old plants accumulate higher amounts of plant defense compounds over a longer growth period; another possible hypothesis could be that young plants need to primarily allocate their resources to their growth process [2,22,23]. Nevertheless, regardless of the plant's physiological explanation, each kind of leaf contains relatively high levels of flavonoids and terpenoids, but the older leaves did contain significantly more.

Nevertheless, it is not possible to imagine a very precise selection of leaves during a harvest at a real industrial scale. Thus, we decided to continue the study only based on a representative mix of leaf ages. The next samples were composed of young and mature leaves. Additionally, they also contained a proportion of old leaves that may fit with what can be really collected in fields by local farmers.

*3.2. Evaluation of the Infusions of Herbal Mixes*

3.2.1. Choice of the Mixture Ratios and Phytochemical Analysis of the Infusions

Herbal mixtures constitute an important part of ethnopharmacological research. Thus, numerous herbal blends have been shown, alone or in combination, to be able to improve health [24]. *C. operculatus* and *M. arvensis* could be considered a potentially interesting combination of individual herbs. Therefore, we tested the potential health benefits of a herbal blend comprising the dried leaves of *C. operculatus* and *M. arvensis*.

Before oven drying, the leaves of *C. operculatus* and *M. arvensis* were preincubated in a stainless-steel jar at 60 °C for 2 h. This incubation process corresponds to a blanching process. Blanching was used as a preliminary step of drying to obtain a high-quality dried herbal drink. It inhibits enzymatic activities. This thermal incubation process also negatively affects the chlorophyll content [25]. However, it leads to lower yields of secondary metabolites in the herb [26]. Furthermore, if many herbal drinks are found to have health benefits for their consumption, they (preferentially) need to have (at least a rather) good taste. From that point of view, our incubated herbal leaves were more preferred in aroma than the nonincubated leaves (data not shown). The incubation process also yielded better results in terms of the taste and aroma of *C. operculatus* leaves compared with nonincubated leaves. The aroma of herbal plants is due to the presence of volatile compounds that easily stimulate the sense of smell [27].

The yields of the extracted flavonoids and terpenoids from hot-water-infused mixtures of *C. operculatus* and *M. arvensis* in various proportions (Mix 1 to 5 corresponding to *C. operculatus* ratios of 100/90/70/50/0%, respectively) were measured (Table 4). The total concentrations of each family of compounds considerably varied between the different proportions of the herbal blends. The values of flavonoids range from 0.76 mg QE/g to 1.39 mg QE/g. All the mixtures have lower total flavonoid content when compared with the *C. operculatus* leaf infusions. This decrease in the flavonoid content of the infusions prepared from the herb mixture could be partially attributed to the addition of *M. arvensis*, which has a lower concentration of flavonoid compounds. Nevertheless, the different levels did not directly change as the ratio changed. Thus, the mixed infusions of the two plant species did not react as simple mixtures. Other explanations should be explored to explain the difference between the theoretical values of pure mixtures and the measured results obtained with the real extracts.

**Table 4.** Flavonoid and terpenoid contents in the different infusions prepared on dried leaves of *C. operculatus* and *M. arvensis* mixed in various proportions.

| Phytochemicals | Treatment | | | | |
|---|---|---|---|---|---|
| | Mix 1 | Mix 2 | Mix 3 | Mix 4 | Mix 5 |
| Flavonoids (mg QE/g) | 1.39 [a] ± 0.025 | 1.02 [c] ± 0.029 | 0.99 [d] ± 0.02 | 0.76 [e] ± 0.11 | 1.28 [b] ± 0.015 |
| Terpenoids (mg SE/g) | 13.12 [d] ± 0.26 | 12.8 [d] ± 0.204 | 14.31 [c] ± 0.162 | 15.36 [b] ± 0.243 | 26.36 [a] ± 0.292 |

Results expressed as mean value ± standard deviation (SD); Mix 1: 100% *C. operculatus*; Mix 2: 90% *C. operculatus*, 10% *M. arvensis*; Mix 3: 70% *C. operculatus*, 30% *M. arvensis*; Mix 4: 50% *C. operculatus*, 50% *M. arvensis*; and Mix 5: 100% *M. arvensis*. Means (of three replicates) followed by a different letter [a–e] are significantly different at an alpha level of 0.05% according to a Least Significant Difference test.

The values for terpenoids range from 12.8 mg SE/g to 26.36 mg SE/g. The terpenoid content is high in mint leaves. On the other hand, the leaves of *C. operculatus* also contain terpenoids. Therefore, the terpenoid content is high in all the infused mixtures with various proportions.

3.2.2. Sensory Analysis of the Different Infusions

The five samples placed in the herbal blend bags were put into separate flasks. Freshly boiled mineral water (modalities 100 mL and 150 mL) was poured into 2 g of each herbal

blend sample and infused for 10 min. The sensory properties of the herbal blend beverage were tested by 45 respondents. The most accepted proportion of the herbal ingredients was chosen using a nine-point hedonic scale.

Compared with the initial 100 mL reference (2 g DW of powder infused in 100 mL of hot water), a lower level of bitterness was noted with 150 mL, and the best blend was Mix 3. Each sample of leaf powder comprised a mix of old, medium, and young leaves.

As hedonic scales are useful for testing and measuring preferences [28], a nine-point hedonic scale was used to determine acceptability. The sensory characteristics of the drink comprising *C. operculatus* and *M. arvensis* in various proportions are presented in Table 5. All the *C. operculatus* and *M. arvensis* drinks showed a brownish-to-yellow color. *M. arvensis* yields a specific flavor (i.e., metabolites) with refreshing effects. *C. operculatus* alone has a bitter taste (if too concentrated) but a specific, unique, and subtle aroma if the concentration is not too high. According to the analysis of the *C. operculatus* leaves prepared using this method, the specific unique aroma and taste of *C. operculatus* could be maintained. Compared with the reference conditions (infusion of 2 g of dried powder in 100 mL of hot water), one tea bag (of 2 g) mixed with 150 mL of water led to better results. For brewing in 150 mL of water, the nine-point hedonic scale sensory analysis of the participants' responses showed overall acceptability, ranging from 4.9 to 8.5 points, which meant that the responses ranged from neither like nor dislike to extremely like (Table 5). It was found that Mix 3 received the highest score, with a hedonic value of 8.5 (the maximum score is 9 for the sensory profiling). Therefore, the most appropriate proportion of the herbal blend beverage was 70% *C. operculatus* and 30% *M. arvensis*.

**Table 5.** Hedonic scaling evaluation of herbal blend beverage.

| Sample/Water | Mix 1 | Mix 2 | Mix 3 | Mix 4 | Mix 5 |
|---|---|---|---|---|---|
| 1/100 | 5.1 [bc] $\pm$ 1.53 | 5.8 [abc] $\pm$ 1.64 | **6.4 [a] $\pm$ 1.63** | 6.1 [ab] $\pm$ 1.61 | 4.8 [c] $\pm$ 1.85 |
| 1/150 | 4.9 [b] $\pm$ 1.16 | 5.4 [b] $\pm$ 1.18 | **8.5 [a] $\pm$ 0.64** | 6.2 [b] $\pm$ 1.47 | 5 [b] $\pm$ 1.60 |

Values are expressed as mean (*n* = 45) $\pm$ standard deviation (SD). Means followed by a different letter [a–c] are significantly different at an alpha level of 0.05% according to a Least Significant Difference test. Mix 1: 100% *C. operculatus*; Mix 2: 90% *C. operculatus*, 10% *M. arvensis*; Mix 3: 70% *C. operculatus*, 30% *M. arvensis*; Mix 4: 50% *C. operculatus*, 50% *M. arvensis*; and Mix 5: 100% *M. arvensis*. Score 1 = extremely dislike; 2 = dislike very much; 3 = moderately dislike; 4 = slightly dislike; 5 = neither like nor dislike; 6 = slightly like; 7 = moderately like; 8 = like very much; and 9 = extremely like.

As far as an infusion presenting a sensory evaluation below 6 or 7 will not have a chance to be accepted by consumers, it became obvious that only Mix 3 could be of interest. Then, we decided to evaluate further characteristics of such a mix, and only on Mix 3.

### 3.2.3. Effect of Brewing Time on Total Flavonoids and Terpenoids of Mix 3

In order to evaluate the effect of brewing time on phytochemical extraction, an experiment was conducted with brewing times of 6, 10, and 60 min. The total flavonoid and terpenoid contents were measured on Mix 3 infused in 100 mL of boiled water, and the results are listed in Table 6.

**Table 6.** Effect of brewing time on total flavonoid and terpenoid contents of Mix 3 infusions.

| Phytochemicals | Time (min) | | |
|---|---|---|---|
| | **6** | **10** | **60** |
| Flavonoids (mg QE/g) | 1.02 [b] $\pm$ 0.18 | **1.17 [b] $\pm$ 0.009** | 1.54 [a] $\pm$ 0.19 |
| Terpenoids (mg SE/g) | 12.47 [c] $\pm$ 0.16 | **14.41 [a] $\pm$ 0.16** | 13.68 [b] $\pm$ 0.17 |

Values are presented as mean $\pm$ SD. Means followed by a different letter [a–c] are significantly different at an alpha level of 0.05% according to a Least Significant Difference test.

In terms of the total flavonoids, the brewing time of 60 min leads to an increase in the phytochemical content (Table 6). Their average concentration is 1.54 mg QE/g of dry weight (Table 6). A comparison of the flavonoid contents of the herbal blend brewed in 6 min, 10 min, and 60 min showed that the content increased with longer brewing time. This is somewhat logical and in accordance with the previous results of Saklar et al. [29]. The longer duration of brewing provided a longer time of contact between the brewing water and the herbal blend. The extraction process improved, and the total flavonoids were increased because flavonoids are water-soluble compounds. However, somewhat different results were observed for terpenoids. The herbal blend brewed in 60 min exhibited a significantly lower terpenoid content. The content seemed to increase at first and then started to decrease after 10 min of brewing. In our experimental design, the maximum value was obtained at 10 min and reached a value of 14.4 mg/g of dry weight. With 10 min of brewing, the *C. operculatus* and *M. arvensis* drink showed a brownish-to-yellow color. It exerted a natural refreshing mint flavor while maintaining the slightly bitter taste and specific unique aroma of *C. operculatus*. The herbal blend prepared with the 10 min brewing time received the highest score (8 points). The combination of phytochemicals and sensory analysis allowed us to determine that the best brewing time for the herbal blend beverage was 10 min. The most appropriate herbal blend was 2 g of *C. operculatus* and *M. arvensis*, with proportions of 70% and 30%, respectively, and a brewing time of 10 min in 100 mL and (better in) 150 mL of boiling water. The levels of quercetin and menthol were investigated using high-performance liquid chromatography (HPLC) and gas chromatography (GC). The amounts of quercetin and menthol in the samples are 25 mg/100 g and 32 mg/100 g, respectively.

Although several different methods have been developed to evaluate the antioxidant activity of biological samples, it is relatively difficult to measure each antioxidant component separately. Therefore, this study explored a new therapeutic agent of plant origin, namely *C. operculatus* leaves, and attempted to confirm its biological efficacy, including its lipid peroxidation inhibitory efficacy, in various in vitro antioxidant models.

### 3.2.4. Determination of DPPH Radical Scavenging Activity of Mix 3

Flavonoid and terpenoid contents function as protective antioxidants at various levels. The antioxidant activities of these plant bioactive compounds have been widely proven within in vitro tests. The antioxidant activity of the herbal blend was measured with the DPPH free radical scavenging method, and their scavenging activity was compared with that of the standard antioxidant ascorbic acid. The DPPH free radical scavenging activity of the herbal blend and ascorbic acid is shown in Table 7. Each leaf powder sample was a mix of old, medium, and young leaves and corresponded to Mix 3.

**Table 7.** DPPH radical scavenging activity of the lyophilized dried matter from the infusion of the herbal blend.

| Samples | IC$_{50}$ (μg/mL) |
| --- | --- |
| Herbal blend Mix 3 | 35.65 ± 0.12 |
| Ascorbic acid | 5.11 ± 0.02 |

The results in Table 7 show that the DPPH scavenging activity of the herbal blend Mix 3 is seven times weaker than the positive control ascorbic acid. Most medicinal plants have antioxidant activity, although less than that of ascorbic acid, which is generally used as the reference for antioxidant activity. Secondary metabolites can prevent free radicals that cause oxidation [30]; thus, the free radical formation is stable and reduces tissue damage.

### 3.2.5. Lipid Peroxidation Inhibitory Activity of Herbal Blend Mix 3

The ability to inhibit lipid peroxidation is commonly determined by measuring the content of malonyl dialdehyde (MDA), which is a product of the peroxidation process af-

fecting cell membrane lipids. In our experiment (Table 8), MDA reacted with thiobarbituric acid to form a trimethin complex (pink color) with a maximum absorption peak at 532 nm wavelength.

**Table 8.** Lipid peroxidation inhibitory activity of the lyophilized dried matter from the infusion of the herbal blend.

| Samples | IC$_{50}$ (μg/mL) |
|---|---|
| Herbal blend Mix 3 | 6.01 ± 0.75 |
| Trolox | 27.85 ± 1.22 |

In the presented measurements of lipid peroxidation inhibitory activity, the IC$_{50}$ value of the herbal blend is 6.01 ± 0.75 μg/mL. The herbal blend Mix 3 thus appears to be 4.5 more efficient than the pure Trolox molecule used as the reference, revealing a high potential level of brain cell protection induced by the optimized *C. operculatus* × *M. arvensis* infusion. Thus, these results justify further studies that include human response analysis and also allow the continuous promotion of the use of this health-promoting beverage.

**4. Conclusions**

*Cleistocalyx operculatus* leaves contain various compounds, such as flavonoids, saponins, phenols, terpenoids, and alkaloids. The total flavonoid and terpenoid contents of leaves were high. For old leaves, they also were slightly higher than those of the young and mature leaves, which are those mainly and efficiently collected in the field. The herbal blend of *C. operculatus* mixed with 30% of *Mentha arvensis* leaves kept most of the initial specific unique aroma and taste and revealed a DPPH radical scavenging IC50 value of 35.65 (± 0.12) μg/mL and a lipid peroxidation inhibition IC50 value of 6.01 (± 075) μg/mL. However, the addition of *Mentha arvensis* in a 7/3 ratio allowed a reduction in bitterness, which was not so much appreciated, mainly by the young participants of the hedonic test.

This study reveals the potential of the combination of *C. operculatus* (70%) and *M. arvensis* (30%) as a natural drink appreciated by the panel of 19–23 years old consumers. This improved formulation (7/3 ratio) shows potential therapeutic benefits, more specifically, with a reduction in lipid peroxidation and brain function protection. We can conclude that this work should sustain the consumption of *Cleistocalyx operculatus* and may thus contribute to health protection.

**Author Contributions:** Conceptualization, T.T.L.M.; Methodology, T.T.L.M., L.T.B.K., S.T.T.M., D.L.B.N., L.T.B.T., N.T.Q., T.T.A., L.V.H., N.V.P., C.T.M.D. and N.H.M.; Validation, T.T.L.M. and G.E.; Formal analysis, T.T.L.M.; Investigation, T.T.L.M., L.T.B.K., S.T.T.M., D.L.B.N., L.T.B.T., N.T.Q., T.T.A., L.V.H., N.V.P., C.T.M.D. and N.H.M.; Resources, T.T.L.M.; Writing—Original draft, T.T.L.M. and G.E.; Writing—Review & editing, T.T.L.M. and G.E.; Visualization, T.T.L.M.; Supervision, T.T.L.M. and G.E.; Project administration, T.T.L.M.; Funding acquisition, T.T.L.M. All authors have read and agreed to the published version of the manuscript.

**Funding:** This research was funded by NONG LAM University under grant No. CS-CB21-CNSH-03.

**Institutional Review Board Statement:** This research was approved by the National Vietnamese authorities with the approv-al No. 20210108NLU.

**Informed Consent Statement:** Not applicable.

**Data Availability Statement:** Data can be transmitted on demand.

**Acknowledgments:** The authors thank the Management of NONG LAM University and UMR transfrontalière INRAE 1158 BioEcoAgro BIOPI-UPJV for their encouragement. Thanks to his research was funded by NONG LAM University under grant No. CS-CB21-CNSH-03.

**Conflicts of Interest:** The authors declare no conflict of interest.

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
