# Peer review of "Addition of Mentha arvensis in Infusions of Cleistocalyx operculatus Improves the Hedonic Score and Retains the High Antioxidant and Anti Lipid-Peroxidation Effects"

_applsci, doi:10.3390/app13052873_

Round 1

Reviewer 1 Report

In this study, a combination of Cleistocalyx operculatus and  Mentha arvensis leaves was studied in order to exploit the potential use of the resulting mixed infusions as functional food drinks.

Authors’ affiliation: The authors’ affiliations should be aligned using one font and size in accordance with the authors guidelines.

Abstract: Currently the authors only provided the guidelines in presenting the abstract. The authors should present the abstract based on the current study.  

Usage of scientific names: Cleistocalyx operculatus and  Mentha arvensis have been used in the entire document. In its first mention Cleistocalyx operculatus should be written as  the full scientific name is  Cleistocalyx operculatus (Roxb.) Merr. & L.M.Perry, and in subsequent uses, the genus can be abbreviated using the first initial and a period e.g  C.  operculatus. Mentha arvensis should be written as Mentha arvensis L. In subsequent uses, the genus can be abbreviated using the first initial and a period. E.g. M. arvensis.

Table 3: The authors should a footnote indicating that “Means followed by a different letter are significantly different at an alpha level of 0.05” according to an LSD test.

Table 4: The authors should a footnote indicating that “Means followed by a different letter are significantly different at an alpha level of 0.05” according to an LSD test.

Table 5: The authors should a footnote indicating that “Means followed by a different letter are significantly different at an alpha level of 0.05” according to an LSD test.

Table 6: The authors should a footnote indicating that “Means followed by a different letter are significantly different at an alpha level of 0.05” according to an LSD test.

References: The authors used first and surnames on the listed references.

References should include the following details:

1.        Author 1, A.B.; Author 2, C.D. Title of the article. Abbreviated Journal Name Year, Volume, page range.

2.        Author 1, A.; Author 2, B. Title of the chapter. In Book Title, 2nd ed.; Editor 1, A., Editor 2, B., Eds.; Publisher: Publisher Location, Country, 2007; Volume 3, pp. 154–196.

3.        Author 1, A.; Author 2, B. Book Title, 3rd ed.; Publisher: Publisher Location, Country, 2008; pp. 154–196.

4.        Author 1, A.B.; Author 2, C. Title of Unpublished Work. Abbreviated Journal Name year, phrase indicating stage of publication (submitted; accepted; in press).

5.        Author 1, A.B. (University, City, State, Country); Author 2, C. (Institute, City, State, Country). Personal communication, 2012.

6.        Author 1, A.B.; Author 2, C.D.; Author 3, E.F. Title of Presentation. In Proceedings of the Name of the Conference, Location of Conference, Country, Date of Conference (Day Month Year).

7.        Author 1, A.B. Title of Thesis. Level of Thesis, Degree-Granting University, Location of University, Date of Completion.

8.        Title of Site. Available online: URL (accessed on Day Month Year).

Author Response

Dear Reviewer 1, as demanded for the questions, we made some corrections-improvements:

Authors’ affiliation: The authors’ affiliations should be aligned using one font and size in accordance with the authors guidelines.

-the mistake observed within the presentation of the affiliations has been corrected

Abstract: Currently the authors only provided the guidelines in presenting the abstract. The authors should present the abstract based on the current study.

-the apparent mistake associated with the abstract is now corrected (in fact, no mistake appeared to us on the word & PDF documents; this was probably a mistake associated with the MDPI Susy website!?).

Usage of scientific names: Cleistocalyx operculatus and Mentha arvensis have been used in the entire document. In its first mention Cleistocalyx operculatus should be written as the full scientific name is Cleistocalyx operculatus (Roxb.) Merr. & L.M.Perry, and in subsequent uses, the genus can be abbreviated using the first initial and a period e.g C. operculatus. Mentha arvensis should be written as Mentha arvensis L. In subsequent uses, the genus can be abbreviated using the first initial and a period. E.g. M. arvensis.

All these changes have been applied.

Table 3: The authors should a footnote indicating that “Means followed by a different letter are significantly different at an alpha level of 0.05” according to an LSD test.

Done! Thank you very much.

Table 4: The authors should a footnote indicating that “Means followed by a different letter are significantly different at an alpha level of 0.05” according to an LSD test.

Done! Thank you very much.

Table 5: The authors should a footnote indicating that “Means followed by a different letter are significantly different at an alpha level of 0.05” according to an LSD test.

Done! Thank you very much.

Table 6: The authors should a footnote indicating that “Means followed by a different letter are significantly different at an alpha level of 0.05” according to an LSD test.

Done! Thank you very much.

References: The authors used first and surnames on the listed references.

We checked and the corrections have been done.

Reviewer 2 Report

1.Total flavonoid and terpenoid content was determined for the young leaves, mature leaves and old leaves. „The flavonoid and terpenoid contents in the old leaves’ extracts were higher than those measured for young and mature leaves.”

Because there are such large differences, I think the study should have been conducted, for greater accuracy, by leaf category (young leaves, mature leaves and old leaves). It is not justifiable to create only a mixture between the three categories of leaves.

2.„2.2. Preliminary phytochemical screenin” must be replaced with „2.2. Preliminary phytochemical screening”

3.Although the result of the sensory analysis indicated mix 3 as having the highest score, it is not justified to carry out further analyses, respectively

3.2.3. Effect of brewing time on total flavonoids and terpenoids of Mix 3

3.2.4. Determination of DPPH radical scavenging activity of Mix 3

3.2.5. Lipid peroxidation inhibitory activity of Herbal blend Mix 3

for this sample only. I believe that analyzes 3.2.3, 3.2.4 and 3.2.5 should have been performed for all samples considered (Mix 1: 100% 373 Cleistocalyx operculatus; Mix 2: 90% Cleistocalyx operculatus, 10% Mentha 374 arvensis; Mix 3: 70% Cleistocalyx operculatus, 30% Mentha arvensis; Mix 4: 50% 375 Cleistocalyx operculatus, 50% Mentha arvensis; Mix 5: 100% Mentha arvensis).

This was necessary for the correct interpretation of the results and to have an overview of the research carried out.

Author Response

Dear Reviewer 2, as demanded in the questions/comments, we made some corrections/improvements as follows:

  1. Total flavonoid and terpenoid content was determined for the young leaves, mature leaves and old leaves. „The flavonoid and terpenoid contents in the old leaves’ extracts were higher than those measured for young and mature leaves.”

Because there are such large differences, I think the study should have been conducted, for greater accuracy, by leaf category (young leaves, mature leaves and old leaves). It is not justifiable to create only a mixture between the three categories of leaves.

Thank you for this comment. Nevertheless, the study was conducted in order to try to find a quality compromise between “healthy phytochemical content” and “development of an appetence” for the infusion (i.e. good taste for the market first and secondly confirmation of the health potential). We now know that old leaves contain more chemical of interest (at least for certain classes of compounds) but they also contribute to a bad taste. Then, in order to clarify this point, we have here added the following sentence:

“Nevertheless, it is not possible to imagine a very precise selection of leaves during an harvest at real industrial scale. Thus, we decided to continue the study only based on a representative mix of leaf ages. The next samples were composed of young and mature leaves. And they also contained a proportion of old leaves that may fit with what can be really collected in fields by local farmers.”

We hope that it could be more easily understood and that it may fit with the reviewer’s demand. And this allows to keep the results about leaf age, instead of eliminating these indicative -but rather interesting- data.

  1. „2.2. Preliminary phytochemical screenin” must be replaced with „2.2. Preliminary phytochemical screening”

This has been corrected. 

  1. Although the result of the sensory analysis indicated mix 3 as having the highest score, it is not justified to carry out further analyses, respectively

3.2.3. Effect of brewing time on total flavonoids and terpenoids of Mix 3

3.2.4. Determination of DPPH radical scavenging activity of Mix 3

3.2.5. Lipid peroxidation inhibitory activity of Herbal blend Mix 3

for this sample only. I believe that analyzes 3.2.3, 3.2.4 and 3.2.5 should have been performed for all samples considered (Mix 1: 100% 373 Cleistocalyx operculatus; Mix 2: 90% Cleistocalyx operculatus, 10% Mentha 374 arvensis; Mix 3: 70% Cleistocalyx operculatus, 30% Mentha arvensis; Mix 4: 50% 375 Cleistocalyx operculatus, 50% Mentha arvensis; Mix 5: 100% Mentha arvensis).

This was necessary for the correct interpretation of the results and to have an overview of the research carried out.

Thank you for the remark. It is right to consider that all the mixes should have been tested to obtain a full overview of the phytochemical composition and properties. Nevertheless, in terms of beverage development (which is the final goal: “objectivation of a healthy beverage based on Cleistocalix”), only Mix3 reached an organoleptic level high enough to wish a further development. Then, in order to clarify this point of view, we now added the following sentence:

“As far as an infusion presenting a sensory evaluation below 6 or 7 will not have chance to be accepted by consumers, it became obvious that only Mix3 could be of interest. Then, we decided to evaluate further characteristics of such a mix, and only on Mix3.”

We hope that this additional information will sufficiently fit with the demand of reviewer2.

Also, to be sure that what we wanted to show was clearly (better) established since the beginning of the text, we also modified the last sentence of the abstract which is now:

“The optimized 7/3 ratio of the Cleistocalyx/Mentha mix demonstrated potential health benefits and led to the best results in terms of taste approval among those participating in the sensory tests performed in this study”. The idea of balance between health benefits and taste is no more presented. And thus, the lack of chemical evaluation of infusion with bad taste should not more be considered as a main problem. We hope reviewer2 will share our point of view.

Also, in the conclusion, the text is now:

The total flavonoid and terpenoid contents of leaves were high. For old leaves they also were slightly higher than those of the young and mature leaf which are the leaves mainly and efficiently collected at field.”

Reviewer 3 Report

The abstract is missing from the main manuscript. Please resubmit the manuscript with the abstract for further consideration.

Author Response

Dear Reviewer 3, we did not understand why you couldn't see the abstract which was present in the word version that we could download after the first evaluation of the manuscript. Sorry for this. We hope that you will now be able to see-read it. We also made the corrections-improvements demanded by the two other reviewers (see the joined cover letter please) and the details for your comment are also joined below:

  1. The abstract is missing from the main manuscript. Please resubmit the manuscript with the abstract for further consideration.

We do not understand were occurred the mistake, may be when we downloaded the article for the submission. We do see the abstract on the document recorded on the Susy-MDPI system. Sorry to the reviewer to have him made losing his time with a text impossible to evaluate correctly. Sorry. It should be clear now.

For the editor’s information: at least two other mistakes appeared also (degradation of the format of affiliations and inadequate association of two paragraphs)

Round 2

Reviewer 1 Report

I would like to thank the authors for addressing my comments. I hope the authors found this useful. 

Author Response

Dear reviewer, Thank you for your consideration and the time you spent for helping us to improve our submitted article.

Best regards

Eric Gontier (for all the authors)

Reviewer 2 Report

1. OK.

2. OK.

3. a) Regarding the sensory analysis, the authors presented only the average of the samples, without specifying the characteristics that were evaluated (eg appearance, color, taste, smell, etc.). They thus greatly simplified the results...

While I don't dispute the results obtained, perhaps if there were other people involved, the results would have been different. For this reason, I believe that all the samples used should have been analyzed:

-          Determination of DPPH radical scavenging activity;

-          Lipid peroxidation inhibitory activity.

From the data presented in table 5, it can be seen that at least sample 4 also has a good score in the sensory analysis...

b) From the data presented in table 5, it can be seen that Mix 3 has a higher score in the sensory analysis for the dilution Sample/water = 1/150.

What was the reason why the authors analyzed the total content of p=flavonoids and terpenoids for Mix 3 infused in 100 ml of water?

c) I recommend the authors to repeat the experiment, as follows:

- Specifying the characteristics evaluated in the sensory analysis;

- Specification of the considered dilution, which should be in accordance with the results of the sensory analysis;

- Analysis of all samples taken in work, respectively:

-          Determination of DPPH radical scavenging activity;

-          Lipid peroxidation inhibitory activity.

Author Response

Dear Reviewer 2,

please find the detailed modifications in response to your comments.

Reviewer 2

  1. OK.
  2. OK.
  3.  
  4. a) Regarding the sensory analysis, the authors presented only the average of the samples, without specifying the characteristics that were evaluated (eg appearance, color, taste, smell, etc.). They thus greatly simplified the results...While I don't dispute the results obtained, perhaps if there were

other people involved, the results would have been different.

Dear reviewer, thank you for this comment.

Yes, it is right that old people (more than 80, having a degraded taste perception and longtime uses and habits) do appreciate better the taste of Cleistocalyx operculatus infusion than youngers. Nevertheless, only very few of them do consume such infusions because of its relatively high amertume. No younger people do use it because of its bitter taste.

The simple addition of sucrose could be useful to mitigate the bitter taste but we do not want to promote sugar consumption and further risks of developing type2 diabete, here. Then, the chosen option was to mix Cleistocalyx with Mentha. And it worked. The taste was improved and the antioxidant properties were comparable. The anti lipid-peroxidation effects were measured and were significant.

We dispose of pictures about the color of the different infusions as far as pictures of the different leaves but we did not think that it was necessary to present a set of pictures (in the text or as supplementary data)because only hedonic data are presented.

Also, hedonic test is not a full organoleptic test. It is a “preference” test based on previous works from psychological sciences. In the Hedonic scaling, here, evaluators give a mark to express what they feel when drinking the product. The protocol is established/inspired as described in: Lim J., 2011. Hedonic scaling: A review of methods and theory. Food Quality and Preference. 22, 733-747. https://doi.org/10.1016/j.foodqual.2011.05.008 .

The different testers must not be influenced one by each other or one by the supervisor who will influence the tester when giving him technical indications about how to evaluate objective characteristics (color, odor, …components of taste.

In order to fit with your comment, the titled of table5 is now: “Table 5. Hedonic scaling evaluation of herbal blend beverage”, instead of “sensory evaluation…” that led to confusion about what is really measured and how it is measured.

For this reason, I believe that all the samples used should have been analyzed:

- Determination of DPPH radical scavenging activity;

- Lipid peroxidation inhibitory activity.

Dear reviewer, we totally agree with your point of view.

The problem highlighted here is due to a bad writing of the title and a poor clarity of the objectives as presented in the first version of the text.

To be clear, our initial idea-objective was (briefly) as follows:

-Cleistocalix has previously been presented as an interesting healthy plant.

-Nevertheless, if the taste of its infusion is delicate, it is also associated with a bitter taste, in which amertume leads to a rejection of its consumption as such (high relative level of rejection but quite total for the youngest people-consumers).

-In that study, we here first (and only) checked the biochemical properties (composition and activities) of infusions of C. operculatus alone (in order to corroborate previous published results, or not).

-Then, we tried to improve the taste by testing different mixes with Mentha.

-After that, on the best mix (mix 3: 70% C. operculatus / 30% M. arvensis), we checked the chemical composition, anti-oxidant activity and performed trials on mice brains (only focused on the best mix3 for ethical reasons; the other mixes not being of interest at all based on taste).

So, the objective of taste improvement was the main limitation to overcome. Nevertheless, our first writing did not sufficiently and efficiently show that objective.

Then, we have changed the proposed title, we have rewritten the abstract, and we have clarified the gap in the introduction. Some minor (but important changes have been done as necessary all along the text also, including the conclusion). The document is available to see all the corrections.

The title is now: “Addition of Mentha arvensis in infusions of Cleistocalyx operculatus improves the hedonic score and retains the high antioxidant and anti lipid-peroxidation effects.”

The abstract is revised to emphasize the objectives using “Context / Problem / Methods / Results / Conclusion” as subtitles (that can be eliminated or not in the final version (if accepted for publication)).

Also, in the introduction, the “gap” is now written as:

“Nevertheless, a natural amertume (bitterness) is limiting the appetence for the bitter infusions of C. operculatus. Also, no study has been reported on the inhibitory effect of C. operculatus bitter infusions, including the combination of C. operculatus and M. arvensis dried leaves against lipid peroxidation in the mouse brain.”

And finally, the introduction is closed with:

“In this study, a combination of C. operculatus and M. arvensis leaves was studied in order to exploit the potential use of the resulting mixed infusions as functional food drinks. In the herbal blend tested, the total flavonoid and terpenoid concentrations were determined. Sensory tests were also performed to improve the bitter taste of the infusions.  The DPPH radical scavenging activity and the lipid peroxidation inhibitory activity were then determined on the most appetizing ratio of C. operculatus and M. arvensis.”

We hope that our reply to your comment will be able to convince you that, of course, for pure phytochemical analysis many mice should have been sacrificed with no further benefit because the taste of the infusion would not allow it to be further consumed by people.

Whatever will be your final opinion, we really thank you for this critical analysis that allowed us to improve the first submitted version of the article.

From the data presented in table 5, it can be seen that at least sample 4 also has a good score in the sensory analysis...

  1. b) From the data presented in table 5, it can be seen that Mix 3 has a higher score in the sensory analysis for the dilution Sample/water = 1/150.

Sorry, we do not see that result in table5. All hedonic score below 8 will lead to a rejection of the product because of its taste (whatever could be the color or the smell). Would you eventually speak about table4 about phytochemical? If so, the main objective is now more clearly emphasized: improving the taste and at least keeping the good level of phytochemical properties. Thank you.

  1. c) I recommend the authors to repeat the experiment, as follows:

- Specifying the characteristics evaluated in the sensory analysis;

- Specification of the considered dilution, which should be in accordance with the results of the sensory analysis;

- Analysis of all samples taken in work, respectively:

- Determination of DPPH radical scavenging activity;

- Lipid peroxidation inhibitory activity.

Dear reviewer. Thank you very much for this recommendation. We really understand that your experimental plan would allow to get a full design of the experimental data in that way. Nevertheless, as presented above, because the first and main goal of the study was to improve the taste of Cleistocalix operculatus infusion, that had the reputation to be a healthy drink, our objectives are reached at that point. The taste is improved. The main phytochemical characteristics are not degraded. Redoing the whole experiment would allow to get au full set of data but we can not redo it because of time, of money, of people available and more, for ethical reasons, in order not to sacrifice more animals for further experiments that would only partially justify that research.

We hope you will agree with us. Sorry not to fully go in your way, as far as the suggestion was legitimated by the way we initially presented (wrote) the objectives in the first version. We made some changes in order to clarify better what is the aim of that study (details above in response of 3.a; and new title, abstract, introduction, precisions in the text and conclusion).

As a conclusion, we want to thank you again for the critical review of our submitted article and for your time and serious involvement in that reviewing. This allowed us to improve the document. We hope this paper could be accepted in Applied Sciences MDPI but whatever could be the decision, the document has been improved with your help.

Thanks again.

Best regards

Eric Gontier

Reviewer 3 Report

The authors describe the augmentation of antioxidant and anti-lipid peroxidation activity of the mixture of Cleistocalyx operculatus and Mentha arvensis. The author selected herbal blend mix 3 for the DPPH and lipid peroxidation inhibitory assays. To show the augmented activity of the mixture of 70% Cleistocalyx operculatus and 30% Mentha arvensis was used. The authors should provide the results of the activity of the extract of 70% Cleistocalyx operculatus and 30% Mentha arvensis when performed separately to show the enhanced activity of the mixture. The manuscript contains numerous typographical errors. The manuscript should be carefully revised to address such errors.

Abstract: Please briefly describe the objective of the study.

L22: here? Please complete the sentence.

L31: Please rephrase the sentence. Mentha arvensis L..?

Introduction: Please briefly describe the purpose of the study and why the authors decide to use the mixture of these two plants to enhance the activity.

Table 1 is not necessary for the introduction. You can use table 1 in the result and discussion part.

L93: Please use g for gram. Similarly, use mL, min, and h for milliliter, minute, and hour, respectively.  

L103: Is this the correct way to write HoChiMinh City?

L176, 190: 0,22 µm?

L178: Are you sure about the 5 mm particle size? I think it should be 5 µm. Please confirm.

L179: Please mention the dimension of the column in mm not in cm. 4.6 x 250 mm?

Please describe the analytic HPLC method in detail with the percentage of MeCN used over time (isocratic, gradient).

L205: 2 g?

Could you please mention the average age of young, mature, and old leaves?

L267: C.o? use space.

Tables 3, 4, 5, and 6: Please carefully revise.

0.05”? LSD (write full form)? abcde?

L398: Briefly explain what hedonic value is.

436: M.a?

L489: I think brackets are not necessary.

References: Please carefully revise the references and use the ACS format.

Author Response

Dear Reviewer 3,

Please find below, the modifications corresponding to your comments:

Reviewer 3

The authors describe the augmentation of antioxidant and antilipid peroxidation activity of the mixture of Cleistocalyx operculatus and Mentha arvensis. The author selected herbal blend mix 3 for the DPPH and lipid peroxidation inhibitory assays. To show the augmented activity of the mixture of 70% Cleistocalyx operculatus and 30% Mentha arvensis was used.

The authors should provide the results of the activity of the extract of 70% Cleistocalyx operculatus and 30% Mentha arvensis when performed separately to show the enhanced activity of the mixture.

and

Abstract: Please briefly describe the objective of the study.

Dear reviewer, thank you for your important remark. We understand your comment which is based on the way we wrote and presented the article. Finally, we access to a misunderstanding.

In fact, our initial idea-objective was (briefly) as follows:

-Cleistocalix has previously been presented as an interesting healthy plant

-Nevertheless, if the taste of its infusion is delicate, it is also associated with a bitter taste, in which amertume lead to a rejection of its consumption as such (high relative level of rejection but quite total for the youngest people-consumers).

-In that study, we here first checked the biochemical properties (composition and activities) of infusions of C. operculatus alone.

-Then, we tried to improve the taste by testing different mixes with Mentha.

-After that, on the best mix (mix 3: 70% C. operculatus/30% M. arvensis), we checked the chemical composition, anti-oxidant activity and performed trials on mice brains (only focused on the best mix3 for ethical reasons; the other mixes not being of interest based on taste).

So, the objective of taste improvement was the main limitation to overcome. Nevertheless, our first writing did not sufficiently and efficiently show that objective.

Then, we have changed the proposed title, we have rewritten the abstract, and we have clarified the gap in the introduction. Some minor (but important changes have been done as necessary all along the text also, including the conclusion). The document is available to see all the corrections.

The title is now: “Addition of Mentha arvensis in infusions of Cleistocalyx operculatus improves the hedonic score and retains the high antioxidant and anti lipid-peroxidation effects.”

The abstract is revised to emphasize the objectives using “Context / Problem / Methods / Results / Conclusion” as subtitles (that can be eliminated or not in the final version (if accepted for publication).

Also, in the introduction, the ”gap” is written as:

“Nevertheless, a natural amertume (bitterness) is limiting the appetence for the bitter infusions of C. operculatus. Also, no study has been reported on the inhibitory effect of C. operculatus bitter infusions, including the combination of C. operculatus and M. arvensis dried leaves against lipid peroxidation in the mouse brain.”

And finally, the introduction is closed with:

“In this study, a combination of C. operculatus and M. arvensis leaves was studied in order to exploit the potential use of the resulting mixed infusions as functional food drinks. In the herbal blend tested, the total flavonoid and terpenoid concentrations were determined. Sensory tests were also performed to improve the bitter taste of the infusions.  The DPPH radical scavenging activity and the lipid peroxidation inhibitory activity were then determined on the most appetizing ratio of C. operculatus and M. arvensis.”

We hope that our reply to your comment will be able to convince you.

We really thank you for this critical analysis that allowed us to improve the first submitted version of the article.

The manuscript contains numerous typographical errors. The manuscript should be carefully revised to address such errors.

The manuscript has been carefully revised. If needed, we can apply it a second time to MDPI English service for a second round of English improvement (first round done under MDPI reference: “[English ID: english-54629]” and certificate below.

L22: here? Please complete the sentence. Done

L31: Please rephrase the sentence. Mentha arvensis L..? Done; Now written “Linné” in order to avoid “L” double point (L..)

Introduction: Please briefly describe the purpose of the study and why the authors decide to use the mixture of these two plants to enhance the activity.

Done. Changes have been done in the writing of the introduction to clarify that point.

Table 1 is not necessary for the introduction. You can use table 1 in the result and discussion part.

Done. Thank you.

L93: Please use g for gram. Similarly, use mL, min, and h for milliliter, minute, and hour, respectively.

Done. Thank you.

L103: Is this the correct way to write HoChiMinh City?

Done. It is now well written as in Vietnamese. Thank you

L176, 190: 0,22 μm? We confirm 0.22µm (as far as it is most often enough with 0.45µm)

L178: Are you sure about the 5 mm particle size? I think it should

be 5 μm. Please confirm.

Done. Yes you are right, sorry.

L179: Please mention the dimension of the column in mm not in

  1. 4.6 x 250 mm?

Done. Thank you.

Please describe the analytic HPLC method in detail with the

percentage of MeCN used over time (isocratic, gradient).

It is now written more clearly as: “The isocratic mobile phase was acetonitrile:water:acetic acid (30:70:1) at a flow rate of 1 mL/min.”. Thank you.

L205: 2 g?

Yes it is really 2g. Thank you

Could you please mention the average age of young, mature, (plantes de 3 ans, young: 1-3 weeks estimated typical length 7-9 cm, mature (4-5 weeks typical length 12-14 cm), old (6 weeks and more typical length 10-12 cm) and old leaves?

Done. It is now clearly indicated: “Thus, we determined the total flavonoid and terpenoid levels in the young/mature/old leaves (respectively 1-3/4-5/>6 weeks old and 7-9/12-14/10-12 cm length) of 3 years old C. operculatus plants”.

L267: C.o? corrected! Thank you.

Tables 3, 4, 5, and 6: Please carefully revise. 0.05”? LSD (write full form)? abcde?

Done. It is now written:” 0.05% according to a Least Significant Difference test.”

L398: Briefly explain what hedonic value is.

 Done. It is now indicated:” (the maximum score is 9 for the sensory profiling).”

436: M.a?

Now written as: “C. operculatus and M. arvensis”

L489: I think brackets are not necessary.

Done. Thank you

References: Please carefully revise the references and use the

ACS format.

Done carefully, thank you

Round 3

Reviewer 2 Report

1. Regarding the sensory analysis, the authors presented only the average of the samples, without specifying the characteristics that were evaluated (eg appearance, color, taste, smell, etc.). They thus greatly simplified the results...While I don't dispute the results obtained, perhaps if there were other people involved, the results would have been different.

”It is right that old people (more than 80, having a degraded taste perception and longtime uses and habits) do appreciate better the taste of Cleistocalyx operculatus infusion than youngers. Nevertheless, only very few of them do consume such infusions because of its relatively high amertume. No younger people do use it because of its bitter taste.”

I disagree. I know at least 5-6 people aged 25-30 who drink coffee or other sugar-free drinks, precisely because they prefer the bitter taste.

The authors only simplified the method of presenting the results, without detailing the evaluated characteristics, as they were specified.

2. From the data presented in table 5, it can be seen that Mix 3 has a higher score in the sensory analysis for the dilution Sample/water = 1/150.

Sorry, we do not see that result in table 5. All hedonic score below 8 will lead to a rejection of the product because of its taste (whatever could be the color or the smell). Would you eventually speak about table 4 about phytochemical? If so, the main objective is now more clearly emphasized: improving the taste and at least keeping the good level of phytochemical properties. Thank you.

I disagree. I know at least 5-6 people aged 25-30 who drink coffee or other sugar-free drinks, precisely because they prefer the bitter taste.

From the data presented in table 5, it can be seen that Mix 3 has a higher score in the sensory analysis for the dilution Sample/water = 1/150.

How do you know that all hedonic score below 8 will lead to a rejection of the product because of its taste (whatever could be the color or the smell). How do you know if you have not properly redone the sensory analysis of the 5 analyzed samples?

3. I recommend the authors to repeat the experiment, as follows:

- Specifying the characteristics evaluated in the sensory analysis;

- Specification of the considered dilution, which should be in accordance with the results of the sensory analysis;

- Analysis of all samples taken in work, respectively:

- Determination of DPPH radical scavenging activity;

- Lipid peroxidation inhibitory activity.

Redoing the whole experiment would allow to get au full set of data but we can not redo it because of time, of money, of people available and more, for ethical reasons, in order not to sacrifice more animals for further experiments that would only partially justify that research.

I understand the reasons why experiments cannot be redone. I warmly recommend the authors to conduct complete experiments from the start in the future.

Author Response

Dear reviewer,

We really consider all the comments you did and that really helped us in improving this submitted article.

I do (Prof Eric Gontier) agree with the fact you mention, that here, only a partial organoleptic evaluation is done. An organoleptic serious approach needs to establish a panel of evaluators. Then, as described in the literature presenting the 9-point hedonic scale, the function is to obtain a non biased evaluation by a (relatively) large panel of consumers (meaning, people with no organoleptic formation) who have to say if they like dislike, much-very much.

Here we performed such an hedonic test. No more. May be we wrote somewhere too much about taste, but it is NOT an organoleptic test.

We understand the fact that if much more would be done (more experiment, replication, more results...) the article would have more weight for the scientific community. But we can not do more now. Even if we could, the article could not be published in a very very first rank journal.

Sorry, I can not find more arguments to try to convince you that , yes of course more results should be good; but yet, there are enough proofs and results to justify our research project.

Sorry for that.

Once again, this difficult situation in which I can not fully answer in the way of one of the reviewers, should not hide the fact that we really thank you very much for the time you spent and for your argumentation that obliged us to more deeply revise our document.

Best regards

Eric Gontier

Reviewer 3 Report

The authors have addressed most of my concerns. However, there are still several typographical errors in the manuscript. The authors should gingerly revise the manuscript to address such typographical errors.

The authors have not followed the ACS format for the references. Could you please follow the ACS format for the references?

Author Response

Dear reviewer,

Thank you very much for your comment.

We do our best to (now) fully fir with the ACS format as described in https://pubs.acs.org/doi/full/10.1021/acsguide.40303

We hope that no mistake remain into the presentation of the references list.

Thank you very much for your consideration.

Best regards

Eric Gontier (for all the authors)